

# No association of *COMT* with insight problem solving in Chinese college students

Xiaolei Yang[1,2], Jinghuan Zhang[1] and Shun Zhang[1]

[1] Department of Psychology, Shandong Normal University, Jinan, Shandong, China
[2] College of Life Science, Qilu Normal University, Jinan, Shandong, China

## ABSTRACT

Genes involved in dopamine (DA) neurotransmission, such as the catechol-O-methyltransferase gene (*COMT*), have been suggested as key genetic candidates that might underlie the genetic basis of insight. In a sample of Chinese college students, this study examined whether *COMT* was associated with individual differences in the ability to solve classic insight problems. The results demonstrated that *COMT* was not associated with insight problem solving and there was no gender-dependent effect. This study, together with previous studies, raises the possibility of a complex relationship between *COMT* and insight problem solving.

# INTRODUCTION

Although recent advancements in neuroscience studies of the insight phenomenon have led to a greater understanding of the brain mechanisms of insight, the genetic correlates underlying these mechanisms remain largely unknown. To explore the genetic correlates of insight, there have been attempts to identify insight-related genes. Because dopamine (DA) and DA-related brain regions (e.g., prefrontal cortex) are implicated in the cognitive processes of insight, genes involved in DA neurotransmission, such as the catechol-O-methyltransferase gene (*COMT*), have been suggested as the key candidate genes.

The *COMT* is mapped to chromosome 22q11 and the enzyme encoded by this gene catalyzes the inactivation of monoaminergic neurotransmitters (DA, adrenalin, and noradrenalin) by extra-neuronal transfer of a methyl group to catechol compounds (*Tenhunen et al., 1994*). Although being widely expressed throughout the brain, *COMT* appears to play a particularly important role in the degradation of DA in the prefrontal cortex (PFC).

Consistent with the role of *COMT* in prefrontal catecholamine degradation, the impact of *COMT* in modulating prefrontal-related cognitive functions has been reported in various studies. For example, genetic variants of *COMT* have been repeatedly implicated in different executive functions and behaviors (e.g., *Bellgrove et al., 2005*; e.g., *Boettiger et al., 2007*; *Bruder et al., 2005*; *Malhotra et al., 2002*; *Paloyelis et al., 2010*; *Smith & Boettiger, 2012*; *Tunbridge et al., 2004*). There has been evidence showing that genetic variants of

Corresponding authors
Jinghuan Zhang,
zhangjinghuan@sdnu.edu.cn
Shun Zhang,
yinxingren1986@hotmail.com

*COMT* are associated with individual differences in behavioral task-related brain activity and resting brain functional connectivity (e.g., *Congdon et al., 2009*; *De Frias et al., 2010*; *Jaspar et al., 2015*; *Markett et al., 2016*; *Mier, Kirsch & Meyer-Lindenberg, 2010*; *Smolka et al., 2005*; *Stokes et al., 2011*; *Tian et al., 2013*; *Tunbridge et al., 2013*; *Winterer et al., 2006*). Moreover, since COMT enzyme activity is influenced by gender (*Chen et al., 2004*), many studies have also shown a gender-dependent effect of *COMT*. For example, *Mione et al. (2015)* and *White et al. (2014)* reported a gender-dependent effect of *COMT* on inhibitory control and related brain activity. *Elton et al. (2017)* demonstrated that *COMT* exerted a gender-dependent effect on resting brain functional connectivity and brain activity during decision-making task.

As for the insight phenomenon, to date, there have been two studies that have investigated the associations of *COMT* with different measures of insight problem solving. *Jiang, Shang & Su (2015)* first explored the associations of seven *COMT* single-nucleotide polymorphisms (SNPs) with individual differences in solving classical insight problems. The results indicated that both the rs4680 and rs4633 polymorphisms were significantly associated with insight problem solving and the rs5993883 polymorphism demonstrated a significant gender-dependent effect, with the association only present in males. Based on these findings, *Han et al. (2018)* further examined whether two of the previously reported *COMT* SNPs (rs4680 and rs5993883) were associated with performance in the Remote Associates Test (RAT; *Mednick, 1962*), which is also commonly used to assess insight problem solving (*Bowden et al., 2005*). However, the results demonstrated that only the rs5993883 polymorphism was associated RAT performance. Because gender-dependent effect was not examined in this study, it is not clear whether the finding about the rs5993883 polymorphism would be consistent with *Jiang, Shang & Su (2015)* study.

These two studies provided important and valuable information concerning the relationship between *COMT* and insight. However, as no validation study has been conducted so far, the findings of these two studies should be viewed as hypothesis generating and need to be further validated in independent samples. In this study, we focused on the ability to solve classic insight problems and examined whether the association of *COMT* with insight problem solving could be validated in a sample of Chinese college students.

## MATERIALS & METHODS

### Participants

Details about the participants have been described in a previous study (*Zhang & Zhang, 2016*). Briefly, the participants were 425 unrelated healthy Han Chinese college students (76.7% females, $M_{age} = 18.9$ years, $SD = 0.84$) without self-reported history of psychiatric disorder. This study was approved by the Institutional Review Board of Shandong Normal University (IRB approval number: sdnudp-061). Written informed consent was obtained from each participant.

### DNA extraction and genotyping

Peripheral venous blood sample was collected from each participant and genomic DNA was extracted using the method described in a previous study (*Zhang & Zhang, 2016*). The SNP

selection was largely based on *Jiang, Shang & Su (2015)* study (rs737865, rs5993883, rs4633, rs6267, rs4818, rs4680). One putative functional SNP (rs6269) at the coding region was also included. Genotyping for all SNPs was performed by using the Sequenom MassARRAY iPLEX system. The genotyping success rate was >99.8%. For quality control, 5% of random DNA samples were genotyped twice for each SNP to calculate the genotyping error. The genotyping accuracy was 100%.

### Insight problems

As previously described (*Zhang & Zhang, 2016*), five verbal insight problems and five figural insight problems were used in this study (see Appendix A). All of these problems fulfilled the criterion of "pure" insight problems since they all necessarily require a reconstructing process for their solution (*Weisberg, 1995*). Example of verbal problems: "Lan and Hong were born on the same day of the same month of the same year to the same mother and the same father—yet they are not twins. How is that possible?" Example of figural problems: "How can you arrange 6 identical pencils in such as way as to form 4 identical triangles whose side areas are all equal, without modifying the pencils in any way?" Verbal and figural problems were presented in a counterbalanced order between participants. The participants were given 2 min to solve each problem and were asked to report whether they had previous knowledge of the problems and the solutions. The average number of familiar problems was 0.34 (SD = .85). Three participants who reported being familiar with all five verbal insight problems were excluded from analyses for verbal insight problems. The accuracy rate was calculated as percentage correct on unfamiliar problems.

### Statistical analysis

Hardy–Weinberg equilibrium was tested by Fisher's exact test using Plink v1.9 software (*Chang et al., 2015*). Single SNP analysis and SNP × gender interaction analysis were performed under the genotypic model using linear regression in Plink v1.9. For the SNP of minor allele homozygotes <5%, minor allele homozygote carriers and heterozygote carriers were collapsed into one group for further analysis. To test the aggregate association signals considering all *COMT* SNPs, gene-based analysis and gene-based gene × gender interaction analysis were performed using the Multi-marker Analysis of GenoMic Annotation (MAGMA) approach (*De Leeuw et al., 2015*). The MAGMA approach is based on a multiple linear principal components (PCs) regression model. By projecting the multivariate linkage disequilibrium (LD) matrix of SNPs in a gene, PCs that explain the genetic variations are first extracted. These PCs are further used as predictors of a phenotype under a linear regression framework to test the association between the gene and the phenotype. Empirical $p$ values ($p_{emp}$) were obtained by using the permutation procedure with 10,000 permutations.

## RESULTS

Table 1 shows the average accuracy rates for total, verbal, and figural insight problems. No significant effect of age or gender was observed. The correlation between the verbal and figural insight problem solving was .32 ($p < .01$). Allele frequencies of the seven *COMT*

**Table 1  Descriptive statistics.**

|  | Age | Accuracy rate | | |
|---|---|---|---|---|
|  |  | Total insight problem solving | Verbal insight problem solving | Figural insight problem solving |
| Total | 18.92 (.84) | 27.5% (.19) | 24.6% (.23) | 30.6% (.24) |
| Male | 19.04 (.98) | 29.9% (.23) | 28.1% (.26) | 31.7% (.27) |
| Female | 18.88 (.80) | 26.8% (.17) | 23.6% (.22) | 30.3% (.23) |

**Table 2  Characteristics of the genotyped *COMT* SNPs.**

| SNP[a] | Position[b] | Location | Allele (minor/major) | MAF (%) | HWE $p$ | | |
|---|---|---|---|---|---|---|---|
|  |  |  |  |  | Total | Male | Female |
| rs737865 | 19942598 | Intron1 | C/T | 29.4 | .907 | 1.00 | .892 |
| rs5993883 | 19950115 | Intron1 | G/T | 41.1 | .841 | .675 | 1.00 |
| rs6269 | 19962429 | Exon3 | G/A | 37.8 | .182 | 1.00 | .124 |
| rs4633 | 19962712 | Exon3 | T/C | 26.4 | .105 | .795 | .089 |
| rs6267 | 19962740 | Exon3 | T/G | 7.4 | .493 | 1.00 | .408 |
| rs4818 | 19963684 | Exon4 | G/C | 37.1 | .253 | .836 | .150 |
| rs4680 | 19963748 | Exon4 | A/G | 26.7 | .062 | 1.00 | .023 |

**Notes.**

HWE, Hardy–Weinberg equilibrium; MAF, minor allele frequency.

[a]SNPs are listed down the column in sequential order from the 5′ end to the 3′ end of the sense strand of *COMT*.

[b]Physical position is based on human genome assembly GRCh38.p12.

SNPs in our sample were similar to those of Han Chinese in the 1000 Genomes Project (Table 2). No significant deviation from the Hardy–Weinberg equilibrium was observed, except for rs4680 which slightly deviated from Hardy–Weinberg equilibrium in females ($p = .023$).

Table 3 summarizes the results of the single SNP analysis and the SNP × gender interaction analysis. No significant association between SNP and insight problem solving was observed and there was no supporting evidence for the SNP × gender interaction. Moreover, the gene-based analysis did not detect any significant association (Table 4).

## DISCUSSION

Prefrontal DA is thought to play a key role in the cognitive processes of creativity. According to the Dual Pathway to Creativity model (*De Dreu, Baas & Nijstad, 2008*; *De Dreu et al., 2012*; *Nijstad et al., 2010*), prefrontal DA is closely related to the convergent processing mode and facilitates creative insight by incremental search and systematic processes of obvious and readily available ideas. Since *COMT* is the main factor controlling prefrontal DA levels, it is reasonable to speculate that genetic variants of *COMT* may be associated with individual differences in solving insight problems.

Previous study has provided evidence for the effect of *COMT* on insight problem solving (*Jiang, Shang & Su, 2015*). To further validate the generality of these findings, this study examined whether *COMT* was associated with individual differences in solving classic insight problems. According to previous study, seven *COMT* SNPs were genotyped and

**Table 3  Results of single SNP analysis and SNP × gender interaction analysis.**

| SNP | Total insight problem solving | | | | Verbal insight problem solving | | | | Figural insight problem solving | | | |
|---|---|---|---|---|---|---|---|---|---|---|---|---|
| | Genotype | | Genotype × Gender | | Genotype | | Genotype × Gender | | Genotype | | Genotype × Gender | |
| | Test statistic | $p_{emp}$ | Test statistic | $p_{emp}$ | Test statistic | $p_{emp}$ | Test statistic | $p_{emp}$ | Test statistic | $p_{emp}$ | Test statistic | $p_{emp}$ |
| rs737865 | 0.421 | .811 | 0.382 | .826 | 2.78 | .251 | 1.24 | .543 | 1.50 | .470 | 0.234 | .889 |
| rs5993883 | 0.793 | .667 | 0.541 | .766 | 0.938 | .632 | 0.653 | .729 | 0.767 | .685 | 1.94 | .374 |
| rs6269 | 0.639 | .726 | 0.343 | .840 | 0.073 | .964 | 0.826 | .658 | 2.19 | .334 | 0.015 | .992 |
| rs4633 | 0.447 | .805 | 3.90 | .146 | 1.60 | .449 | 4.46 | .113 | 0.797 | .672 | 1.59 | .460 |
| rs6267 | −0.359 | .733 | 1.83 | .175 | −0.316 | .748 | 0.350 | .548 | −0.357 | .720 | 2.29 | .133 |
| rs4818 | 0.634 | .726 | 0.241 | .886 | 0.100 | .953 | 0.408 | .806 | 1.89 | .391 | 0.049 | .975 |
| rs4680 | 0.204 | .905 | 2.31 | .315 | 1.84 | .409 | 2.72 | .255 | 0.518 | .774 | 1.13 | .578 |

Notes.
Single SNP effect and SNP × gender interaction were tested under the genotypic model using linear regression with a 2df joint test. For SNP (rs6267) with minor allele homozygotes <5%, minor allele homozygote carriers and heterozygote carriers were collapsed into one group (the dominant model) for analysis. Empirical $p$ values ($p_{emp}$) were obtained by 10,000 permutations.

**Table 4  Results of gene-based analysis and gene-based gene × gender interaction analysis.**

| Total insight problem solving | | | | Verbal insight problem solving | | | | Figural insight problem solving | | | |
|---|---|---|---|---|---|---|---|---|---|---|---|
| COMT | | COMT × Gender | | COMT | | COMT × Gender | | COMT | | COMT × Gender | |
| $Z$[a] | $p_{emp}$ | $Z$[a] | $p_{emp}$ | $Z$[a] | $p_{emp}$ | $Z$[a] | $p_{emp}$ | $Z$[a] | $p_{emp}$ | $Z$[a] | $p_{emp}$ |
| −0.098 | .530 | −0.286 | .602 | −0.106 | .535 | −0.554 | .709 | −0.296 | .612 | −0.145 | .555 |

Notes.
Gene-based analysis and gene-based gene × gender interaction analysis were performed using the MAGMA approach. Empirical $p$ values ($p_{emp}$) were obtained by 10,000 permutations.
[a]The $Z$ value for the gene or gene × gender interaction, based on its $p_{emp}$.

were further tested for their associations with the ability to solve classic verbal and figural insight problems. To maximize the statistical power to detect weak associations, a gene-based analysis was also conducted to test the joint association of all seven SNPs. However, in contrast with previous findings, the results demonstrated that *COMT* was not associated with the ability to solve classic insight problems and there was no gender-dependent effect.

In this study the effect of *COMT* was examined in a sample of Chinese college students, while the sample of *Jiang, Shang & Su (2015)* consisted of Chinese high school students. Although the two samples were homogeneous in ethnic background (both samples were of Han Chinese origin and the studied SNPs showed similar allele frequencies), there were significant differences in terms of age and average accuracy for solving insight problems. It was found that college students had lower average accuracy compared with high school students. The differences in average accuracy may reflect an age-related change in PFC and prefrontal DA functions. PFC is one of the last brain regions to mature during development (*Arain et al., 2013*; *Giedd et al., 1999*; *Sowell et al., 2001*). From adolescence to adulthood, the maturation of PFC is characterized by a relatively increased *COMT* expression and decreased prefrontal DA levels (*Tunbridge, Lane & Harrison, 2007*; *Wahlstrom et al., 2010*). Since prefrontal DA facilitates creative insight, there might be a corresponding decrease

in the ability to solve insight problems in the maturation from adolescents to adults. This may partly explain why college students had lower average accuracy.

Based on the age-related change in PFC and prefrontal DA functions, a number of studies have investigated the influence of *COMT* during development and provided evidence for the age-dependent effect of *COMT* (*Barnett et al., 2007*; *Dumontheil et al., 2011*; *Gothelf et al., 2005*; *Gothelf et al., 2013*; *Meyer et al., 2016*; *Tunbridge, Lane & Harrison, 2007*). Considering the discrepancy between our findings and those by *Jiang, Shang & Su (2015)*, it could be speculated that the effect of *COMT* on insight problem solving might also be age-dependent. The relative increase in *COMT* expression (and the resulting decreased prefrontal DA levels) from adolescence to adulthood may result in changes in the relationships between *COMT* and insight problem solving in high school and college students. Future studies are guaranteed to test this hypothesis.

Another possible explanation for the discrepancy between our findings and those by *Jiang, Shang & Su (2015)* is that *COMT* may interact with other DA-related genes, such as dopamine D2 receptor gene (*DRD2*) and dopamine transporter gene (*DAT1*), to influence insight problem solving. Although insight problem solving, as measured by close-ended response, is thought to rely more heavily on the convergent processing mode, the divergent processing mode is also required to sample potential solutions for the problem. Thus, creative insight most likely originates from the interplay between the two processing modes (*Boot et al., 2018*; *Cropley, 2006*; *Nijstad et al., 2010*). Because the two processing modes are implicated in different DA-related brain regions (convergent processing mode to the PFC and divergent processing mode to the striatum), there is the possibility that genes related to prefrontal DA (e.g., *COMT*) may interact with genes related to striatal DA (e.g., *DRD2* and *DAT1*) to influence insight problem solving. There has been evidence showing that *COMT* interacts with *DRD2* and *DAT1* to influence creative potential and creative achievement (*Zabelina et al., 2016*; *Zhang, Zhang & Zhang, 2014*). Future studies should examine whether gene-gene interactions could explain the discrepancy.

Both this study and the study conducted by *Jiang, Shang & Su (2015)* used classic insight problems as insight tasks. Besides classic insight problems, there have been attempts to examine the association of *COMT* with other insight tasks, such as RAT. In a recent study aimed to explore the genetic correlates of convergent and divergent thinking, *Han et al. (2018)* investigated the association of two *COMT* SNPs (rs4680 and rs5993883) with RAT performance. However, to make things more complicated, this study yielded somewhat different results compared to those obtained from our study and the study conducted by *Jiang, Shang & Su (2015)*. Thus, to determine whether this discrepancy was caused by the use of different insight tasks and to provide a more comprehensive understanding of the relationship between *COMT* and insight, future studies should examine whether *COMT* is related to other insight tasks, such as Matchstick Arithmetic (*Knoblich et al., 1999*) and Rebus Puzzles (*MacGregor & Cunningham, 2008*). Moreover, future studies should also include non-insight problems as control tasks and this would help to determine whether the association reflects the specific genetic contribution to insight problem solving.

Several limitations of this study should be addressed. First, the sample size of this study may not provide adequate power to detect weak associations of the very small effect size.

For single SNP analysis, the statistical power of our sample is sufficient (80% power) to detect effect size ($R^2$) greater than 0.02 (with $\alpha = 0.05$). Thus, for lower effect sizes, type II error cannot be excluded. Because the insight phenomenon, like other complex traits, might be highly polygenic and influenced by thousands of genetic variants with small individual effects (*Gratten et al., 2014*; *Manolio et al., 2009*; *Robinson, Wray & Visscher, 2014*), it is important for future studies to validate these results in larger samples. Second, the imbalanced gender ratio in our sample may bias the results of gender-dependent analysis (*Mione et al., 2015*). Although *COMT* is well known to be sexually dimorphic (*Chen et al., 2004*; *Tunbridge, Lane & Harrison, 2007*), gender-dependent effect was not observed in this study. The small sample size of male participants in our sample did not allow definite conclusions as to whether the effect was gender-dependent or due to the relative lack of male participants. Future studies are necessary to draw a definite conclusion. Third, the relatively small number of insight problems, as well as the low accuracy may result in sparse data, which could also bias the results. In this study, a total of ten insight problems were employed; however, since the average accuracy was low, the data was sparse. To account for the potential bias introduced by the sparse data in the statistical analysis, permutation test procedures were employed to generate empirical $p$ values. Although the permutation test procedures allow for the possibility of sparse data (relaxing the assumptions about normality of continuous data), the presence of a floor effect could not be excluded. Thus, the results of this study should still be treated with caution.

## CONCLUSIONS

In conclusion, in a sample of Chinese college students, this study did not demonstrate evidence for the association of *COMT* with insight problem solving. This study, together with previous studies, raises the possibility of a complex relationship between *COMT* and insight problem solving.

### Funding
This study was supported by the National Natural Science Foundation of China (No. 31470999, 31771235), the MOE (Ministry of Education in China) Project of Humanities and Social Sciences (No. 16YJC190030), the Science and Technology Projects of Shandong (China) (No. ZR2014CQ017, ZR2015CL024), and the Research Center of Qilu Culture (Shandong Normal University, Jinan, China). The funders had no role in study design, data collection and analysis, decision to publish, or preparation of the manuscript.

### Grant Disclosures
The following grant information was disclosed by the authors:
National Natural Science Foundation of China: 31470999, 31771235.
MOE (Ministry of Education in China) Project of Humanities and Social Sciences: 16YJC190030.

Science and Technology Projects of Shandong (China): ZR2014CQ017, ZR2015CL024. Research Center of Qilu Culture (Shandong Normal University, Jinan, China).

## Competing Interests

The authors declare there are no competing interests.

## Author Contributions

- Xiaolei Yang performed the experiments, analyzed the data, contributed reagents/materials/analysis tools, prepared figures and/or tables, authored or reviewed drafts of the paper, approved the final draft.
- Jinghuan Zhang conceived and designed the experiments, contributed reagents/materials/analysis tools, authored or reviewed drafts of the paper, approved the final draft.
- Shun Zhang conceived and designed the experiments, performed the experiments, analyzed the data, contributed reagents/materials/analysis tools, prepared figures and/or tables, authored or reviewed drafts of the paper, approved the final draft.

## Human Ethics

The following information was supplied relating to ethical approvals (i.e., approving body and any reference numbers):

This study was approved by the Shandong Normal University's Institutional Review Board (sdnudp-061).

## Data Availability

The raw data (gender, age, insight test scores, and COMT genotypes) are provided in Data S1.

## Supplemental Information

Supplemental information for this article can be found online at http://dx.doi.org/10.7717/peerj.6755#supplemental-information.

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
