# Peer review of "No association of *COMT* with insight problem solving in Chinese college students"

_PeerJ, doi:10.7717/peerj.6755_

## Round 0.1 · original submission · Major Revisions

Your manuscript has been evaluated by an expert in Neurogenetics and an expert in Cognitive Neuroscience who, in a complementary way, have provided suggestions and comments that will be useful for the revision of the text.

One of the main problems raised is that all sections, in particular both the Introduction and the Methods, require more details.

English language usage. I concur with the comments about the need to improve the English language prose. You may need to retain the services of a professional editing consultant, one who has experience with scientific publications; simply asking a colleague with English language proficiency may not suffice.

·

Basic reporting

The article is well written, although literature references should be implemented to provide adequate field background/context.

Experimental design

no comment

Validity of the findings

A more careful discussion of data is suggested (see "General comments for the author").

Additional comments

In this study Yang and coll. attempt to replicate findings of a recent study (Jiang et al., 2015), reporting an association between the COMT polymorphism and insight problem solving performance. After a concise, although thorough description of their results, which show no association, the authors outline some discrepancies between their study and Jiang’s one.

General comment
Although the functional influence the COMT polymorphism on dopamine clearance at the prefrontal cortex level is likely well known by scientists working in the field of behavioral neuroscience, an overview of this topic is totally missing in the present version of the study.
A brief overview of this topic on the introduction section would clarify the biological rationale underpinning the interest of this study (see Chen JS et al., Am. J. Hum. Genet. 2004; Bellgrove MA Exp Brain Res 2005 studies, among others).
In addition, although the study investigates a specific behavior, i.e. the insight problem solving, a brief overview of previous studies reporting association/no association between the COMT polymorphism and cognitive abilities, with particular reference to those that also investigated sex-dependent influences, should be briefly summarized in the introduction section (see Mione V. et al., Frontiers in Behav Neurosci 2015; Zabelina D.L. et al., PlosOne 2016; Jaspar M. et al., Cerebral Cortex 2015; White TP et al. Neuropsychopharmacology 2014; Elton A et al., Front. Hum. Neurosci 2017).
Of note, the authors of this study do not quote a recent study, which being related to their field of interest they might know, i.e. Han W et al., PeerJ 2018.

This implementation would provide the present study a wider audience.

Specific issues

- It is well known that the last part brain to mature is the prefrontal cortex. The authors should consider the age of participants as a possible cause of the discrepancy between their findings as compared to those of Jiang et al., 2015 study.
Indeed, the Mage is significantly different in the two studies: here18.9, SD=0.84 (Jiang et al: Mage=16.54 years, SD = 0.70).
Considering that the maturation of the prefrontal cortex networks is a late achievement, being completed at 20-22 years of age, the age factor should be carefully discussed, also within the perspective of future studies.

- The authors should comment on the possible influence of the robust imbalance between female (76.7%) and male fractions in the cohort of participants to their study.

- The authors should verify that genotype frequencies within male and female groups were consistent with the Hardy-Weinberg equilibrium using a specific test. The sentence ‘No significant deviation from Hardy-Weinberg equilibrium was observed (data not shown)’ is not sufficient.

Reviewer 2 ·

Basic reporting

1. The introduction is meagre to a fault. Only a bare skimming of the theoretical background in the topic.

2. Methodological standards are questionable.
(a) Important details about the task are missing. The reader is given no indication as to what the task is and how the fit the criterion of being an insight task.

3. Results: Key results are missing (no information provided on how many problems out of the 10 were classified as familiar problems and how many were classified unfamiliar problems.

5. Several grammatical and semantic errors throughout the manuscript.

6. Several instances of #NULL fields in the data where the behaviour results should be.

Experimental design

2. Methodological standards are questionable. For instance, at the level of the task alone there are there severe problems:
(a) Important details about the task are missing. The reader is given no indication as to what the task is and how the fit the criterion of being an insight task.
(b) No non-insight tasks were employed as control tasks.
(c) Only 10 problems (which translates to 10 trials) were used as a whole which is very low.

3. Results: Key results are missing (no information provided on how many problems out of the 10 were classified as familiar problems and how many were classified unfamiliar problems. So it is impossible to tell what exactly the accuracy rate reflects (percentage correct on unfamiliar problems). In addition, the accuracy rate was extremely low (e.g., total insight problem solving for all participants around 27.5%). So the data is very sparse and potentially based on only 1-2 correct responses of the participants.

Validity of the findings

4. The findings are not congruous with the study their are trying to replicate (Jiang et al.). However, it is questionable whether this can be considered a replication study when not even clear that the same materials were used in this study as in the Jiang study. Certainly this study has less insight problems than the Jiang et al. study.

Additional comments

1. The introduction is meagre to a fault. Only a bare skimming of the theoretical background in the topic.

2. Methodological standards are questionable. For instance, at the level of the task alone there are there severe problems:
(a) Important details about the task are missing. The reader is given no indication as to what the task is and how the fit the criterion of being an insight task.
(b) No non-insight tasks were employed as control tasks.
(c) Only 10 problems (which translates to 10 trials) were used as a whole which is very low.

3. Results: Key results are missing (no information provided on how many problems out of the 10 were classified as familiar problems and how many were classified unfamiliar problems. So it is impossible to tell what exactly the accuracy rate reflects (percentage correct on unfamiliar problems). In addition, the accuracy rate was extremely low (e.g., total insight problem solving for all participants around 27.5%). So the data is very sparse and potentially based on only 1-2 correct responses of the participants.

4. The findings are not congruous with the study their are trying to replicate (Jiang et al.). However, it is questionable whether this can be considered a replication study when not even clear that the same materials were used in this study as in the Jiang study. Certainly this study has less insight problems than the Jiang et al. study.

5. Several grammatical and semantic errors throughout the manuscript.

6. Several instances of #NULL fields in the data where the behavioral results should be.

---

## Round 0.2 · accepted · Accept

Thank you for answering in full the comments of both reviewers.

#